# Identifiable Causal Inference with Noisy Treatment and No Side Information

**Antti Pöllänen** *antti.pollanen@aalto.fi*
*Department of Computer Science*
*Aalto University*

**Pekka Marttinen** *pekka.marttinen@aalto.fi*
*Department of Computer Science*
*Aalto University*

**Reviewed on OpenReview:** *https://openreview.net/forum?id=EONPcsEZ2f*

## Abstract

In some causal inference scenarios, the treatment variable is measured inaccurately, for instance in epidemiology or econometrics. Failure to correct for the effect of this measurement error can lead to biased causal effect estimates. Previous research has not studied methods that address this issue from a causal viewpoint while allowing for complex nonlinear dependencies and without assuming access to side information. For such a scenario, this study proposes a model that assumes a continuous treatment variable that is inaccurately measured. Building on existing results for measurement error models, we prove that our model's causal effect estimates are identifiable, even without side information and knowledge of the measurement error variance. Our method relies on a deep latent variable model in which Gaussian conditionals are parameterized by neural networks, and we develop an amortized importance-weighted variational objective for training the model. Empirical results demonstrate the method's good performance with unknown measurement error. More broadly, our work extends the range of applications in which reliable causal inference can be conducted.

## 1 Introduction

Causal inference deals with how a treatment variable $X$ causally affects an outcome $Y$. This is different from just estimating statistical dependencies from observational data, because even when we know that $X$ causes $Y$ and not the other way around, the statistical relationship could be affected by confounders $Z$, i.e., common causes of $X$ and $Y$. Knowing the causal relationships is crucial in fields that seek to make interventions, e.g., medicine or economics (Pearl, 2009; Peters et al., 2017; Imbens & Rubin, 2015).

Causal inference may be complicated by variables being subject to noise, e.g., inaccurate measurement, clerical error, or self-reporting (often called misclassification for categorical variables). If not accounted for, it is well-known that this error can bias statistical and causal estimates in a fairly arbitrary manner, and consequently, measurement error models have been widely studied to address this bias (Carroll et al., 2006; Buonaccorsi, 2010; Schennach, 2016; 2020).

In this paper, we assume the treatment $X$ and outcome $Y$ to be continuous, while the confounders $Z$ can be categorical, discrete or continuous. (The case of a binary or categorical $X$ is briefly discussed in Appendix A.) While measurement error may occur in any of the variables, we explicitly model it only in $X$. We implicitly also include measurement error in $Y$, but in our model it is indistinguishable from the inherent noise in the true value of $Y$. The practical impact of this is limited, since the measurement error for $Y$ does not bias regression due to its zero-mean additive nature that we assume. On the other hand, measurement error in $Z$ could be relevant (Miles et al., 2018), but it is restricted outside the scope of this work.

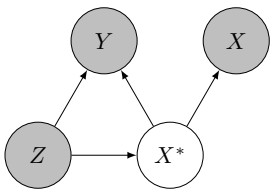

Figure 1: Causal graph for our proposed model. The observed variables $X$ (noisy treatment), $Y$ (effect/outcome), and $Z$ (confounders) are shaded to distinguish them from the hidden variable $X^*$ (true treatment).

The causal graph expressing our assumptions is depicted in Figure 1. An example reflecting this scenario in the context of healthcare is the effect of blood pressure on some continuous health outcome $Y$, such as arterial stiffness as measured by pulse wave velocity, or kidney function as measured by glomerular filtration rate (GFR) or by the concentration of albumin in urine. We assume $X$ is a single measurement of blood pressure, and thus it is highly subject to instantaneous variation, which plays the role of the measurement error. Some additional error probably also results from measurement apparatus inaccuracy. Here, the blood pressure that actually matters in terms of health outcomes (and whose effects we are interested in) is a longer term sliding window average, which is unobserved and which we denote by the latent variable $X^*$.

To enable statistical identifiability, we assume the measurement error to be additive, zero-mean and independent from the other variables. These are a standard set of assumptions in the measurement error literature, called strongly classical measurement error. Non-classical measurement error is also widely studied and well documented to occur in practice (see Schennach (2016) for a review), but it is restricted outside the scope of this work. However, we believe our modeling and inference methodology is general enough that extensions to these cases are fairly straightforward. Figure 2 presents a synthetic example of the skewing effect of measurement error in the treatment. While it is often believed that this type of measurement error only introduces attenuating bias to a naive estimate not accounting for the error (Yi et al., 2021), this example demonstrates that on the contrary, amplification is also possible. Here attenuation happens only around $X^* = 0$, but for small and large $X^*$ the naive estimate amplifies the strength of the dependency.

The measurement error problem is often addressed by making additional measurements to enable model identification. The information obtained in this way could be a known measurement error variance, or so-called side information, such as repeated measurements, instrumental variables, or a gold-standard sample of accurate measurements (Yi et al., 2021). However, we study the scenario where none of this is available, and we must rely on assumptions that could reasonably be made a priori, such as the error being additive and having a zero mean. Further, we study the scenario where the dependencies between the variables can be complex and nonlinear, and an observed confounder is present. There is a gap in the literature for causal inference when all of these challenging aspects are present.

To address this gap, our paper provides a solution by inferring a structural causal model (SCM) from observational data using deep latent variable modeling and importance weighted variational inference (Zhang et al., 2019). The inferred SCM enables the computation of any interventional and counterfactual distributions on the variables in the model, but to evaluate the fit, we consider the accuracy of the estimation of the following quantities: 1) the function $\mu_Y(z, x^*) = \mathbb{E}[Y|z, do(x^*)]$, 2) the noise variances $\tau^2$ and $\sigma^2$, and 3) the function $p(y|do(x^*))$, which maps $x^*$ to the density of $Y|do(x^*)$. The combination of 1) and 2) also evaluates the estimation accuracy of $p(y|z, do(x^*))$, since with $Z$ satisfying the backdoor criterion and by assuming a conditionally Gaussian $Y$, we have $p(y|z, do(x^*)) = p(y|z, x^*) = \mathcal{N}(y|\mu_Y(z, x^*), \sigma^2)$. These parameters/distributions were chosen because they are either model parameters, reflect probable use cases of the model, or both. To facilitate practical computation, we assume conditionally Gaussian variables. However, we emphasize that the purpose of this is to simplify computations, and the identification of the model does not rely on this assumption, see Section 2.3.1 for the details. We leave up to future work to relax it by using flexible probability density estimators such as normalizing flows. We evaluate our algorithm on a wide variety of synthetic datasets, as well as semi-synthetic data.

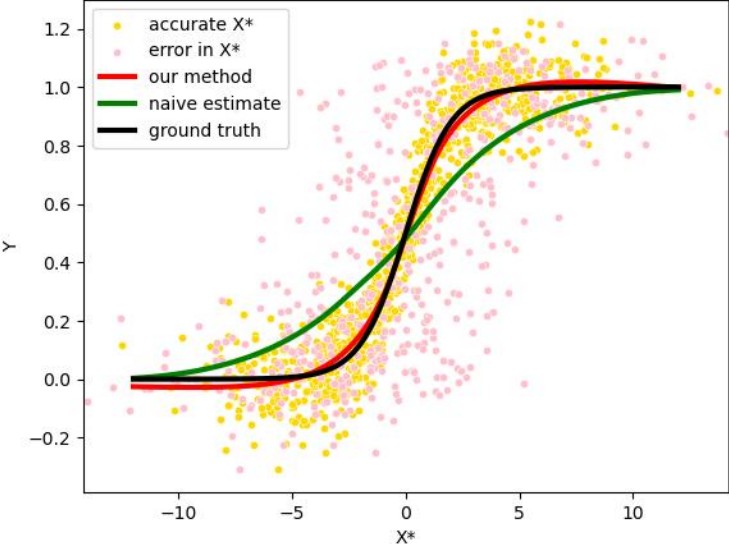

Figure 2: Comparison of our method (CEME) against a naive method which does not account for measurement error in the treatment $X^*$. The accurate values of $X^*$ are hidden from both methods. Ground truth is the true mean function of the data generating process. The same data are displayed both with and without measurement error in $X^*$. It can be seen that our method "CEME" fits the error-free data (even if they are not seen by any method) whereas the "naive" method fits the data with error and cannot estimate the true regression function accurately. A similar example is presented by Zhu et al. (2022).

There is previous literature on all of the challenges tackled in this work, although not previously studied all together. For an overview of the literature on the bias introduced by measurement error in causal estimation, see Yi et al. (2021). This literature includes qualitative analysis by encoding assumptions of the error mechanism into a causal graph (Hernán & Robins, 2021), and quantitative analysis of noise in the treatment (Zhu et al., 2022), outcome (Shu & Yi, 2019), confounders (Pearl, 2012; Miles et al., 2018) and mediators (Valeri & Vanderweele, 2014). However, most of this literature focuses on simple parametric models, such as the linear model, and the identifiability of the causal effect is achieved through a fully known error mechanism or side information, such as repeated measurements, instrumental variables, or a gold standard sample of accurate measurements. Exceptions include Miles et al. (2018) which studies confounder measurement error with no side information and a semiparametric model, Hu et al. (2022) which similarly to us uses deep latent variable models and variational inference, but assumes known measurement error variance, and Zhu et al. (2022) which studies treatment measurement error with a nonparametric model and notably even allowing for unobserved confounding, but utilizes both repeated measurements and an instrumental variable as side information. Schennach & Hu (2013) use non-parametric sieve estimation with no side information, without considering covariates $Z$ that could increase complexity significantly. We take them into account not only to increase the accuracy of predictions, but crucially to allow adjusting for confounders to obtain causal effects. Concurrently to us, Gao et al. (2024) study a similar model to ours and also use variational inference. However, they do not include a theoretical analysis of identifiability as we do. See Appendix A for additional references to identifiability results on models related to ours.

To summarize, this work contributes to the literature by successfully inferring causal effects in a setting with a novel combination of challenging aspects: measurement error, no side information, no knowledge of the measurement error variance, and complex nonlinear dependencies including dependency on confounders that must be adjusted for in order to estimate the causal effect. To show the identifiability of our model, we extend a previous result by Schennach & Hu (2013) to allow for confounders/covariates, and adjust it to the specific assumptions of our model. Finally, this work contributes by bridging different disciplines: it highlights that measurement error models constitute SCMs and uses machine learning, specifically variational inference, as a foundation for inferring causal effects.

## 2   Methods

### 2.1   Models for causal estimation

For causal inference, this study uses the structural causal model (SCM) framework (Pearl, 2009). A SCM consists of the following components: 1) A list of endogenous random variables $V = (V_1, V_2, \ldots, V_N)$. 2) A directed acyclic graph $G = (V, E)$ called the causal graph, whose vertices are the variables $V$ and whose edges $E$ express causal effects between variables. 3) A list of exogenous random variables $U = (U_1, U_2, \ldots, U_N)$ that represent external noise and are mutually independent. 4) A list of deterministic functions $F = (f_1, f_2, \ldots, f_N)$, called the causal mechanisms, such that for $i = 1, \ldots, N$ we have $V_i = f_i(PA_i, U_i)$, where $PA_i$ are the random variables that are the parents of $V_i$ in $G$. This construct allows not only the modeling of observational data but also the effects of interventions. An intervention on variable $V_i$ in an SCM is defined as a replacement of the mechanism $f_i$ with another mechanism $\tilde{f}_i$. The simplest case is a hard intervention, denoted by $do(V_i = v_i)$, where $\tilde{f}_i$ simply assigns a constant value $v_i$ to $V_i$. When estimating any causal quantities from data, we assume the data to be i.i.d. according to the SCM. An intervention applied on one individual or unit (each corresponding to a data point) does not affect others. The properties of consistency and not having multiple versions of an intervention (Hernán & Robins, 2021) follow trivially from the definition of an SCM.

To model measurement error, we use an SCM with the causal graph $G$ depicted in Figure 1. Note that the causal graph is also a Bayesian network that implies conditional independencies according to d-separation (Bishop, 2006). The variables in the graph are the confounders/covariates $Z$, the accurate treatment value $X^*$ which is unobserved, the noisy measurement of the treatment, denoted by $X$, and the noisy value of the outcome, denoted by $Y$. Except for $X^*$, the other variables are observed. The SCM with independent noise terms and no hidden confounders implies the common assumption of causal sufficiency. Therefore, when the SCM is identified, any interventional or counterfactual distributions can be computed. These include e.g. $Y_{x^*}$, which denotes the distribution of $Y$ after intervening with $do(X^* = x^*)$ (alternatively denoted by $Y|do(X^* = x^*)$), or the distribution of $Y_{x_1^*} - Y_{x_2^*}$, which compares the effects of two treatment values $x_1^*$ and $x_2^*$, or the distribution of $Y_{x^*}|Y = y, Z = z$, which denotes the outcome that would have been obtained from intervention $do(X^* = x^*)$ when in reality no intervention was made and the values $Y = y$ and $Z = z$ were observed.

This paper combines themes of statistical and causal estimation, which use related but different definitions of identifiability. However, we use everywhere the same unified definition:

**Definition 1** (Identifiability of an estimand $f$). *Let $\{P_{B,L}^\theta\}_{\theta \in \Theta}$ be a family of joint probability distributions of the observed variables $B$ and latent variables $L$, parameterized by $\theta$. Denote by $P_B^\theta$ the corresponding marginal distribution for $B$. Then an estimand $f(\theta)$ (where $f$ is a deterministic function) is identifiable if for every $\theta, \theta' \in \Theta$ we have*

$$P_B^\theta = P_B^{\theta'} \Rightarrow f(\theta) = f(\theta').$$

Here $f$ represents anything one is interested in estimating from data (including but not limited to causal parameters), and as a special case when $f(\theta) = \theta$, we get the definition of *model identifiability*. Intuitively, these mean that knowing the distribution of the observed variables, for example by having estimated it with an infinite amount of i.i.d. samples, the estimand $f(\theta)$ (or simply $\theta$) can be uniquely determined. In causal inference literature based on SCMs, identifiability often means that an interventional distribution can be computed from infinite observational data and the causal graph structure (Peters et al., 2017). In contrast, with our definition this computation can additionally use any assumptions made about the underlying SCM.

Because $X^*$ is hidden, the SCM or any causal queries that consider the effects of an intervention $do(X^* = x^*)$ are not identifiable without further assumptions, as the relationship between $X^*$ and the other variables can not be estimated. Therefore, for identification, we rely on additional assumptions on the measurement error mechanism $X = f_X(X^*, U_X)$ such that the SCM becomes statistically identifiable. In the literature, much attention has been devoted to making the model identifiable using a known error variance or side information. However, we instead rely on the assumptions of independence and zero-mean additive errors (Schennach & Hu, 2013): First, we assume that the measurement error is *strongly classical*, i.e., the observed treatment $X$

is generated by a causal mechanism

$$X = X^* + \Delta X, \tag{1}$$

where the measurement error $\Delta X$, which acts as the exogenous variable $U_X$, is independent of all other variables except $X$ and has a zero mean. Similarly, we assume for the outcome $Y$ that

$$Y = \mu_Y(Z, X^*) + \Delta Y, \tag{2}$$

where the noise $\Delta Y$ is the exogenous variable $U_Y$ independent of all other variables except $Y$ and has a zero mean. The conditional mean of $Y$, $\mu_Y(z, x^*) = \mathbb{E}[Y|Z = z, X^* = x^*]$, is a deterministic, possibly nonlinear function that is continuously differentiable everywhere with respect to $x^*$. (Note that by uppercase letters we denote random variables and by lowercase letters their specific realized values. Thus $\mu_Y(z, x^*)$ refers to the value of $\mu_Y$ with argument values $z$ and $x^*$, while $\mu_Y(Z, X^*)$ denotes the random variable that is obtained by applying $\mu_Y$ to the random variables $Z$ and $X^*$.) Although not necessary for identifiability (see Section 2.3.1 for details), we further assume Gaussian distributions to facilitate practical computations, yielding

$$X^*|Z \sim \mathbb{N}(\mu_{X^*}(Z), \sigma_{X^*}^2(Z)) \tag{3}$$

$$\Delta X \sim \mathbb{N}(0, \tau^2), \tag{4}$$

$$\Delta Y \sim \mathbb{N}(0, \sigma^2), \tag{5}$$

We assume that $X$, $Y$, and $X^*$ are real-valued scalars, although multivariate extensions are relatively straightforward. The covariate $Z$ can be a combination of categorical, discrete, or continuous variables, but the other variables are assumed to be continuous. Consequently, the learnable parameters are the variance parameters $\tau^2$ and $\sigma^2$ as well as the (deterministic) functions $\mu_{X^*}$, $\sigma_{X^*}$ and $\mu_Y$, which are all parameterized by neural networks to allow flexible dependencies between the variables. The zero-mean assumptions in Equations (4) and (5) are made to achieve statistical identifiability. Otherwise, for any non-zero mean $\mu_{\Delta Y}$, we could obtain a new observationally equivalent model by using $\mu'_Y(z, x^*) = \mu_Y(z, x^*) - \mu_{\Delta Y}$. Similarly, for any non-zero mean $\mu_{\Delta X}$, we could obtain a new observationally equivalent model by using $\mu'_{X^*}(z) = \mu_{X^*}(z) - \mu_{\Delta X}$ and $\mu'_Y(z, x^*) = \mu_Y(z, x^* + \mu_{\Delta X})$.

Finally, we define the following model variants that we evaluate against each other:

**CEME**: Causal Effect estimation with Measurement Error. This is the model defined in Equations (1)–(5). The mean and standard deviation functions $\mu_{X^*}(Z)$, $\sigma_{X^*}(Z)$ and $\mu_Y(Z, X^*)$ are fully connected neural networks.

**CEME$^+$**: Causal Effect estimation with Measurement Error with known noise. This is otherwise the same as CEME, except that the variance of the measurement error $\tau^2$ is assumed to be known.

**Oracle**: This method assumes the model in Figure 3a, where the true treatment $X^*$ is observed and is used directly for fitting the model. Hence, this model provides a loose upper bound on the performance that can be achieved by the models CEME and CEME$^+$ which only observe the noisy treatment $X$. The model consists of only one neural network, parametrizing $\mu_Y(Z, X^*)$.

**Naive**: This is the same as Oracle, except that it naively uses the observed noisy treatment $X$ instead of the true treatment $X^*$ when estimating the causal effect (model depicted in Figure 3b). Hence, it provides a baseline that corresponds to the usual approach of neglecting the measurement error in causal estimation. The model consists of only one neural network, parametrizing $\mu_Y(Z, X)$.

To summarize the roles of the methods, CEME is our proposed method, CEME$^+$ enables comparison with the case of a known measurement error variance $\tau^2$, Oracle is a loose upper bound for performance, and Naive is a baseline. For inference, CEME and CEME$^+$ use amortized variational inference to estimate the latent variable (described in the next section). In contrast, Oracle and Naive, which do not have latent variables, are trained using gradient descent with mean squared error (MSE) loss for predicting $Y$.

## 2.2 Inference

Amortized variational inference (AVI) (Zhang et al., 2019) can be used to estimate the parameters of deep latent variable models (LVMs). The method includes one or more latent variables (true treatment $X^*$ in our

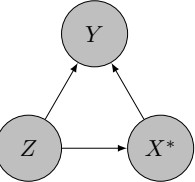
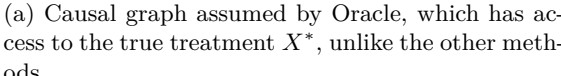
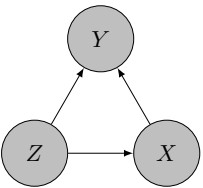

(a) Causal graph assumed by Oracle, which has access to the true treatment $X^*$, unlike the other methods.

(b) Causal graph (incorrectly) assumed by the method Naive.

model), observed variables whose distributions are modeled ($X$ and $Y$ in our case), and optionally observed variables whose distributions are not modeled but on which the model is conditioned (the covariates $Z$ in this paper). We use this method to infer the SCM, after which causal effect estimates of interest can be calculated from the model parameters by Monte Carlo sampling over observed covariates $Z$.

For the distribution $p(X^*, X, Y|Z)$, the method requires defining a generative model (also called decoder) $p_\theta(x^*, x, y|z)$, where $\theta$ denotes the parameters. In this paper, we use the CEME model defined in Equations (1)–(5) with $\theta$ consisting of the standard deviations $\tau$ and $\sigma$ as well as the functions $\mu_{X^*}(z)$, $\sigma_{X^*}(z)$ and $\mu_Y(z, x^*)$, which are modeled as fully connected neural networks. (Thus in terms of the practical computations, the parameters are not the functions itself but their respective neural network weights.)

AVI also requires modeling the posterior $p(X^*|Z, X, Y)$ with a so-called encoder, for which we use $q_\phi(x^*|z, x, y) = \mathbb{N}\left(x^*|\mu_q(z, x, y; \phi), \sigma_q(z, x, y; \phi)\right)$, where $\phi$ denotes the encoder parameters, which are the functions $\mu_q(z, x, y; \phi)$ and $\sigma_q(z, x, y; \phi)$, or alternatively, the weights of the two fully connected neural networks that model these functions.

The parameters $\theta$ and $\phi$ are optimized with stochastic gradient descent using the negative evidence lower bound (ELBO, defined in the next section) as the loss function. This approach can be used for LVMs that generalize the variational autoencoder (VAE) to more than just one hidden and one observed node (Kingma & Welling, 2019) (CEME/CEME$^+$ uses the Bayesian network in Figure 1).

### 2.2.1 Model training

As our objective function, we use the importance weighted ELBO $\mathcal{L}_k$ (Burda et al., 2016), which is a lower bound for the conditional log-likelihood $p_\theta(x, y|z)$:

$$\log p_\theta(\mathbf{x}, \mathbf{y}|\mathbf{z}) = \sum_{i=1}^{N} \log p_\theta(x_i, y_i|z_i) \tag{6}$$

$$= \sum_{i=1}^{N} \log \mathbb{E}_{q_\phi(\ldots)} \left[ \frac{1}{K} \sum_{j=1}^{K} \frac{p_\theta(x^*_{i,j}, x_i, y_i|z_i)}{q_\phi(x^*_{i,j}|z_i, x_i, y_i)} \right] \tag{7}$$

$$\geq \sum_{i=1}^{N} \mathbb{E}_{q_\phi(\ldots)} \left[ \log \frac{1}{K} \sum_{j=1}^{K} \frac{p_\theta(x^*_{i,j}, x_i, y_i|z_i)}{q_\phi(x^*_{i,j}|z_i, x_i, y_i)} \right] \tag{8}$$

$$:= \mathcal{L}_K, \tag{9}$$

where the expectations are with respect to $q_\phi(x^*_{i,j}|z_i, x_i, y_i)$. For each data point $i$, there are $K$ i.i.d. realizations of the latent variable $x^*$, indexed by $j$. The inequality in (8) is obtained by applying Jensen's inequality. The standard ELBO corresponds to $K = 1$. For practical optimization, the expectation in Equation (8) is estimated by sampling the $K$ realizations of the latent variable $x^*_{i,j}$ once per data point $i$.

We use the importance weighted ELBO instead of the standard ELBO because the latter places a heavy penalty on posterior samples not explained by the decoder, which forces a bias on the decoder to compensate for a misspecified encoder (Kingma & Welling, 2019). Increasing the number of importance samples alleviates this effect, and in the limit of infinite samples, decouples the optimization of the decoder

from that of the encoder. To calculate the gradient of the ELBO, it is standard to use the reparameterization trick, meaning that we sample $x_{i,j}^*|z_i, x_i, y_i \sim q_\phi(x_{i,j}^*|z_i, x_i, y_i)$ as $x_{i,j}^* = \epsilon \cdot \sigma_q(z_i, x_i, y_i; \phi) + \mu_q(z_i, x_i, y_i; \phi)$, where $\epsilon$ follows the standard normal distribution. The importance-weighted ELBO objective is optimized using Adam gradient descent. Further training details are available in Appendix B. The algorithm was implemented in PyTorch, with code available for replicating the experiments at `https://github.com/antti-pollanen/ci_noisy_treatment`.

### 2.3 Identifiability analysis

#### 2.3.1 Identifiability of causal estimation with a noisy treatment

The identifiability proof of the CEME model builds closely on an earlier result by Schennach & Hu (2013), of which we provide an overview here. It is an identifiability result on a slightly different measurement error model, which differs from ours in that it does not include the covariate $Z$, but on the other hand it is more general in that it does not assume that $X^*, \Delta X$ and $\Delta Y$ are (conditionally) Gaussian. The exact form of the result is included in Appendix C. The model includes scalar real-valued random variables $Y, X^*, X, \Delta X$ and $\Delta Y$ related via

$$Y = g(X^*) + \Delta Y \quad \text{and} \quad X = X^* + \Delta X, \tag{10}$$

where only $X$ and $Y$ are observed. We assume that 1) $X^*, \Delta X$ and $\Delta Y$ are mutually independent, 2) $\mathbb{E}[\Delta X] = \mathbb{E}[\Delta Y] = 0$, and 3) some fairly mild regularity conditions. It is shown by Schennach & Hu (2013) that this model is identifiable if the function $g$ in Equation (10) is not of the form

$$g(x^*) = a + b\ln(e^{cx^*} + d), \tag{11}$$

and even if it is, nonidentifiability requires $x^*$ to have a specific distribution, e.g. a Gaussian when $g$ is linear. From this result, we see that the CEME model assumptions of conditionally Gaussian variables (made to simplify computation) do not help with identification, as it brings the model towards the non-identifiable special cases. With these preliminaries, we are now ready to prove the following proposition that establishes the identifiability of the CEME model:

**Proposition 1.** *The measurement error model defined in Equations (1)–(5) is model identifiable if 1) for every $z$, $\mu_Y(z, x^*)$ is continuously differentiable everywhere as a function of $x^*$, 2) for every $z$, the set $\chi = \{x^* : \frac{\partial}{\partial x^*}\mu_Y(z, x^*) = 0\}$ has at most a finite number of elements, and 3) there exists $z$ for which $\mu_Y(z, x^*)$ is not linear in $x^*$ (i.e. of the form $\mu_Y(x^*) = ax^* + b$).*

*Proof.* The full proof is provided in Appendix D, but its outline is the following: First, we show that a restricted version of our model where $z$ can only take one value is identifiable as long as $\mu_Y(z, x^*)$ is not linear in $x^*$. This follows from the identifiability theorem by Schennach & Hu (2013) because its assumptions are satisfied by the restricted version of our model, together with the assumptions of Proposition 1.

Second, we show the identifiability of our full model by looking separately at the values of $z$ for which $\mu_Y(z, x^*)$ is or is not linear in $x^*$. For the model conditioned on the latter type of $z$ (nonlinear cases), identifiability was shown in the first part of the proof. For the remaining linear cases, we use the fact that we already identified $\tau$ and $\sigma$ with the nonlinear cases, which results in a linear-Gaussian model with known errors, which is known to be identifiable (see e.g. Equations (4.9-4.17) in Gustafson (2003)). Thus the entire model is identified. $\qquad\square$

We note that assumption 1) in Proposition 1 is satisfied by the neural networks used in this work, as they are continuously differentiable due to them using the extended linear unit (ELU) activation function. Assumption 2) is true in general and does not hold only in the special case where the derivative of the neural network $\mu_Y(z, x^*)$ is zero with respect to $x^*$ in infinitely many points, which is possible only if the function is constant on an interval. Breaking assumption 3) would require the network $\mu_Y(z, x^*)$ to be linear in $x^*$ for every $z$. Even when this is the case, we still hypothesize that the model is identifiable as long as there are multiple values of the linear slope for different $z$. This is suggested by solving a system of equations similar to Gustafson (2003), which appears to have one or at most two distinct solutions. On another hand, the proof of Proposition 1 shows that if there was no $z$ for which $\mu_Y(z, x)$ was linear in $x$, we would not need

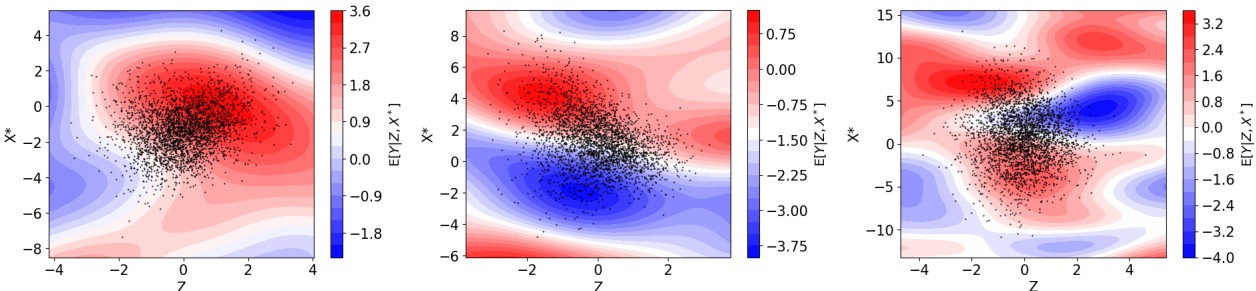

Figure 4: Three realizations of synthetic datasets generated from a Gaussian process with 3000 data points (black dots). The figures show the covariate $Z$ (x-axis), the noiseless versions of the treatment $X^*$ (y-axis), and the outcome $\mathbb{E}[Y|Z, X^*]$ (heatmap).

to assume any restrictions on how $\tau$ and $\sigma$ depend on $Z$, and we surmise that even when there are such $z$, milder restrictions than constant $\tau$ and $\sigma$ would be sufficient. However, we leave a detailed investigation of these cases for future work.

## 3 Experiments and results

We conducted two experiments: One with datasets drawn from Gaussian processes, where a large number of datasets is used to establish the robustness of our results. Second, we use an augmented real-world dataset on the relationship between years of education and wages, to study the effect of complex real-world patterns for which the assumptions of our model do not hold exactly.

### 3.1 Synthetic experiment

#### 3.1.1 Synthetic datasets from Gaussian processes

For the synthetic experiment, we generate datasets whose underlying distribution follows the CEME model with all variables being scalars. Gaussian processes (GPs) are used, see Rasmussen & Williams (2006) for an introduction. In brief, a random function $f(t)$ for $t \in \mathbb{R}^n$ is a Gaussian process if $f(t_1), ..., f(t_k)$ is jointly Gaussian for any $k \in \mathbb{N}$ and any $t_1, ..., t_k \in \mathbb{R}^n$. They are denoted by $\mathrm{GP}(\mu, K)$ and characterized by a mean function $\mu(t_i)$ and kernel function $K(t_i, t_j)$, by which one obtains the mean vector and covariance matrix for the jointly Gaussian $f(t_1), ..., f(t_k)$ (for any $k \in \mathbb{N}$ and any $t_1, ..., t_k \in \mathbb{R}^n$). In this study, the mean functions $\mu(t)$ are constant. The data generation proceeds according to Algorithm 1. For the GPs, we use the squared

---

**Algorithm 1** Generation of synthetic datasets using GPs

1: For $i \in \{1, .., N\}$, sample $z_i$ from $\mathbb{N}(0, 1)$.
2: Sample $\mu_{X^*}$ from $\mathrm{GP}(0, K)$ and $g^{-1} \circ \sigma_{X^*}$ from $\mathrm{GP}(1, K)$. Here $\circ$ denotes function composition and $g(z) = \log(1 + \exp(z))$ is used to ensure the positivity of $\sigma_{X^*}$.
3: For $i \in \{1, .., N\}$, sample $x_i^*$ from $\mathbb{N}(\mu_{X^*}(z_i), \sigma_{X^*}^2(z_i))$.
4: Sample $\mu_Y$ from $\mathrm{GP}((0, 0), K)$.
5: Set $\tau = L\sqrt{\mathrm{Var}[(x_1^*, ..., x_N^*)]}$, where Var denotes unbiased sample variance and $L$ is a constant.
6: Set $\sigma = L\sqrt{\mathrm{Var}[(\mu_Y(z_1, x_1^*), ..., \mu_Y(z_N, x_N^*))]}$.
7: For $i \in \{1, .., N\}$, sample $x_i$ from $\mathbb{N}(x_i^*, \tau^2))$.
8: For $i \in \{1, .., N\}$, sample $y_i$ from $\mathbb{N}(\mu_Y(z_i, x_i^*), \sigma^2)$.

---

exponential kernel $K(u_1, u_2) = \alpha \exp\left(-\frac{|u_1 - u_2|^2}{2l^2}\right)$ with $\alpha = 1$ and lengthscale $l = 2$ and where $|\,.\,|$ denotes the Euclidean norm. Note that for $\mu_{X^*}(z)$ and $\sigma_{X^*}(z)$ the corresponding GPs are over $\mathbb{R}$ while for $\mu_Y(z, x^*)$ the GP is over $\mathbb{R}^2$. To avoid excessive computational cost, the functions $\mu_{X^*}(z)$, $\sigma_{X^*}(z)$ and $\mu_Y(z, x^*)$ are

only approximations of true samples from the GPs (steps 2 and 4 in Algorithm 1), as described in Appendix B along with other experiment details. Three datasets generated in this manner are shown in Figure 4.

The constant $L$ is varied to obtain datasets with different levels of noise, which are either $L = 0.1$ (small noise), $L = 0.2$ (medium noise), or $L = 0.4$ (large noise). The different training dataset sizes used are 1000, 4000, and 16000 data points. The test data (used for evaluating the models) consist of 20000 data points. For each combination of noise level and training dataset size, we generate 200 different datasets (with different underlying distributions) using Algorithm 1. For all of these, we fit each model (CEME, CEME$^+$, Naive and Oracle) 6 times, and for the results pick the run with the best score on a separate validation dataset (distinct from the test dataset). All models use the same neural network architecture to facilitate a fair comparison, i.e. three fully connected hidden layers with 20 nodes each and the ELU activation function.

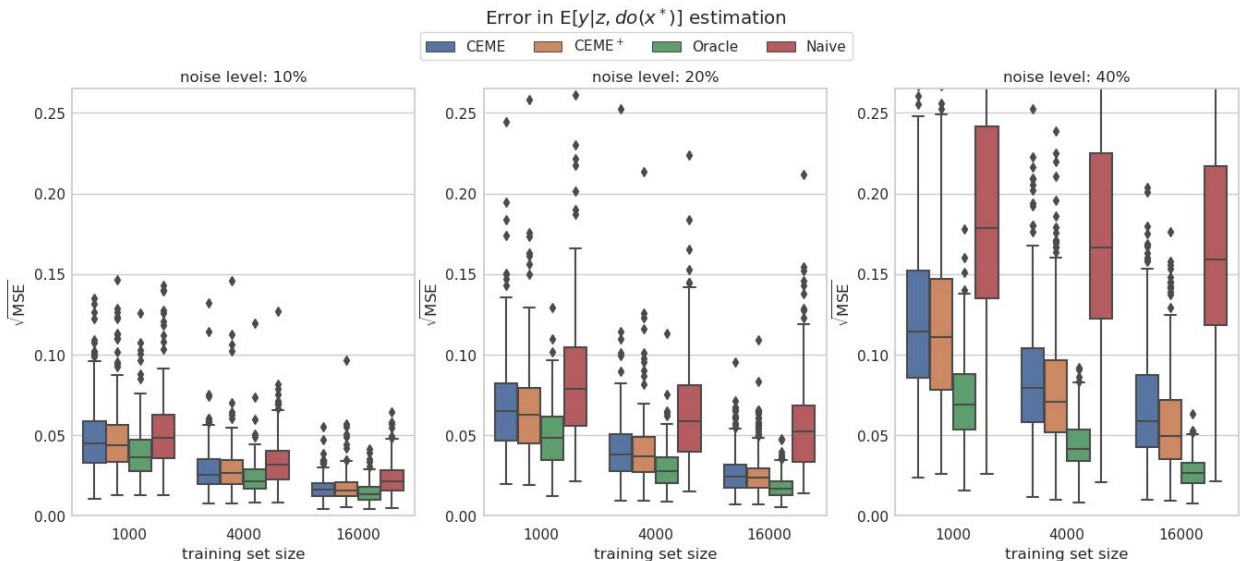

Figure 5: Error in $\mathbb{E}[y|z, do(x^*)]$ estimation in the synthetic experiment. In addition to data for Naive, missing from the figure are some outliers for CEME and CEME$^+$ for noise level 40% and training dataset size 1000. Key observations are that 1) the proposed CEME/CEME$^+$ methods offer clear benefit over Naive that does not account for measurement error, 2) CEME and CEME$^+$ seem to converge with increasing training set size and compare relatively well with the loose upper bound Oracle, and 3) CEME handles unknown measurement error variance well, since CEME$^+$ that knows it, performs only slightly better.

### 3.1.2 Results

The accuracy of the estimation of $\mu_Y(z, x^*) = \mathbb{E}[Y|z, do(x^*)]$ is reported in Figure 5 using the root mean squared error

$$\sqrt{\text{MSE}} = \sqrt{\sum_{i=1}^{N}(\mu_{Y,\theta}(z_i, x_i^*) - \mu_Y(z_i, x_i^*))^2},$$

where $\theta$ denotes the estimate and the sum is over the $N = 20000$ data points in the test set. The accuracy of the estimation of $\sigma$ is reported in Figure 6, which uses the metric relative error $(\sigma_\theta - \sigma)/\sigma$, where $\sigma_\theta$ is the estimate and $\sigma$ the true value. The relative error in the estimation of the measurement error noise $\tau$ is reported in Figure 8 in the same way. The accuracy in the estimation of $p(y|do(x^*))$ is presented in Figure 7 and assessed using *Average Interventional Distance* (AID) Rissanen & Marttinen (2021):

$$\text{AID} = \int p(x^*) \int |p_\theta(y|do(x^*)) - p(y|do(x^*))| \, dy dx^* \tag{12}$$

Here, both the estimated $p_\theta(y|do(x^*))$ and the ground truth $p(y|do(x^*))$ are obtained using the adjustment formula $p(y|do(x^*)) = \int p(y|x^*, z)p(z)dz$ as $Z$ is assumed to satisfy the backdoor criterion (Pearl, 2009).

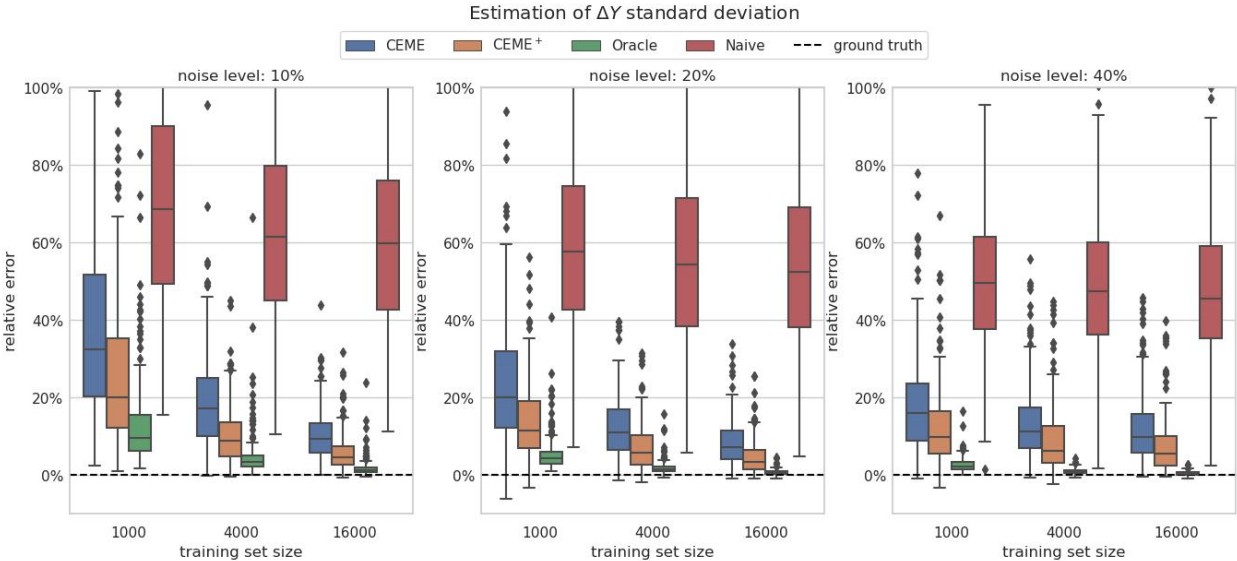

Figure 6: Error in estimation of $\Delta Y$ standard deviation in the synthetic experiment. In addition to data for Naive, missing from the figure are 4 outliers for CEME, 1000 training data and 10% noise as well as 2 outliers for CEME, 4000 training data and 10% noise. Key observations are that 1) the estimates seem to converge for the proposed CEME/CEME$^+$ but not for the baseline Naive, and 2) all the methods tend to overestimate, rather than underestimate, the standard deviation of $\Delta Y$.

All integrals are estimated using Monte Carlo integration, using test data as a sample for $Z$ and $X^*$, and sampling $Y$ uniformly, with the exception that for the ground truth $p(y|do(x^*))$ we integrate over $z$ using the trapezoidal rule.

In these figures, the data are represented as boxplots, where the box corresponds to the interquantile range, and the median is represented as a horizontal bar. The whiskers extend to the furthest point within their maximum length, which is 1.5 times the interquantile range. Outliers beyond the whiskers are represented as singular points. Each data point corresponds to a separate data-generating distribution and the best of six runs based on validation loss.

Overall the CEME algorithms offer significant benefit over not taking measurement error into account at all (algorithm Naive), both in terms of estimating individual causal effect (Figures 5 and 6), and average causal effect (Figure 7). In addition, the results suggest convergence of estimates when increasing dataset size, as expected from the theoretical identifiability analysis. On the other hand, even with knowledge of the true standard deviation (SD) of $\Delta X$, CEME$^+$ does not achieve performance comparable to that of Oracle. This is to be expected as Oracle sees accurate values of $X^*$ for individual data points, which cannot be identified even in principle from information available to CEME and CEME$^+$.

Interestingly, the performance improvement from knowing the true SD of $\Delta X$ (with CEME$^+$) over learning it (with CEME) seems relatively modest. This suggests that identifiability is generally not an issue for CEME, even though the data distributions could be arbitrarily close to the non-identifiable linear-Gaussian case. We also note that all algorithms almost always overestimate the SD of $\Delta Y$, which could be explained by an imperfect regression fit. Furthermore, as detailed in Section 2.2.1, the training of CEME and CEME$^+$ is initialized such that they try to predict $X^*$ close to $X$, which could be another cause for this and also explain why CEME tends to underestimate the SD of $\Delta X$ (Figure 8).

## 3.2 Experiment with education-wage data

We also test CEME with semisynthetic data based on a dataset curated by Card (1995) from data from the National Longitudinal Survey of Young Men (NLSYM), conducted between years 1966 and 1981. The items

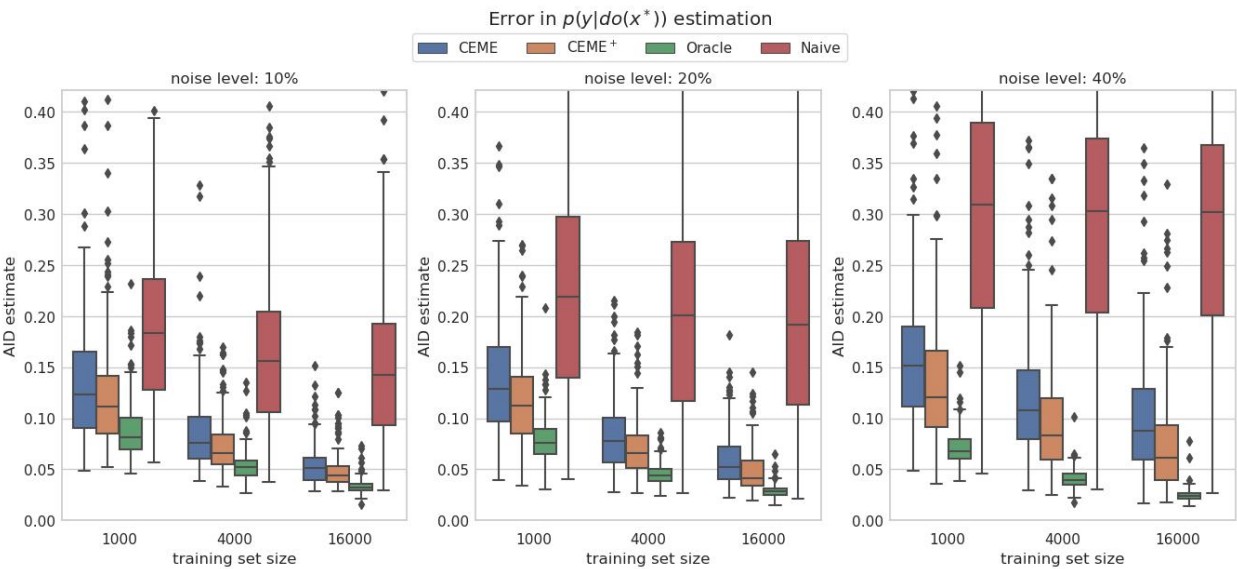

Figure 7: Error in estimation of $p(y|do(x^*))$ in the synthetic experiment. Besides data for Naive, missing from the figure are 3 outliers for CEME and 1 for CEME$^+$, both for 40% noise and 1000 training data. This figure tells a similar story to Figure 5, except that the differences between the methods are more pronounced.

in the dataset correspond to persons whose number of education years is used as treatment $X^*$. As outcome $Y$, we use the logarithm of the wage. The relationship between these is also studied in the original paper (Card, 1995). The dataset also contains multiple covariates, of which this experiment uses a total of 23.

To evaluate our methods, we need to know the ground truth values of the parameters that we estimate. Having access to real observed data alone does not permit this, which is why we augment the real data with synthetic variables that mimic the real ones. The known data generating processes of these synthetic variables enable us to compute the ground truth. To this end, we first train a neural network to predict the outcome, and then modify the dataset by replacing the outcome values with the neural network predictions to which Gaussian noise is added. The noisy treatment $X$ is obtained by adding Gaussian noise to $X^*$. Six separate datasets are created, each corresponding to a different level of SD of $\Delta X$. The levels are proportional to the SD of $X^*$, and are 0%, 20%, 40%, 60%, 80% and 100%. The choice of education years as treatment was in part motivated by the need to know the true treatment accurately to obtain the ground truth causal effect. The CEME/CEME$^+$ models are misspecified for this dataset because the true treatment, the number of education years, is ordinal, instead of conditionally Gaussian, as the models assume. This

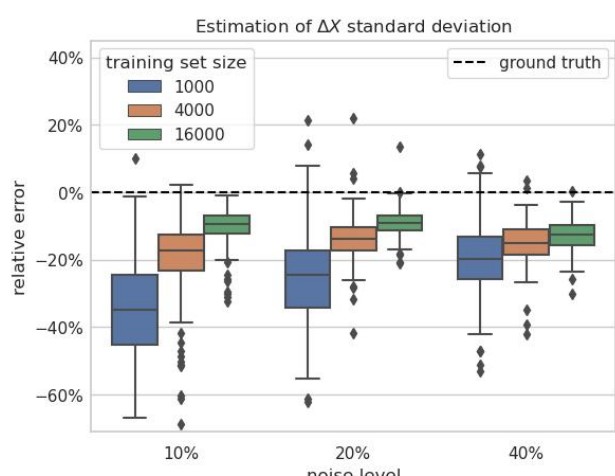

Figure 8: Error in estimation of $\Delta X$ standard deviation by CEME in the synthetic experiment. The estimates seem to converge with increasing training set size.

presents an opportunity to evaluate the sensitivity of the CEME/CEME$^+$ algorithms to model misspecification.

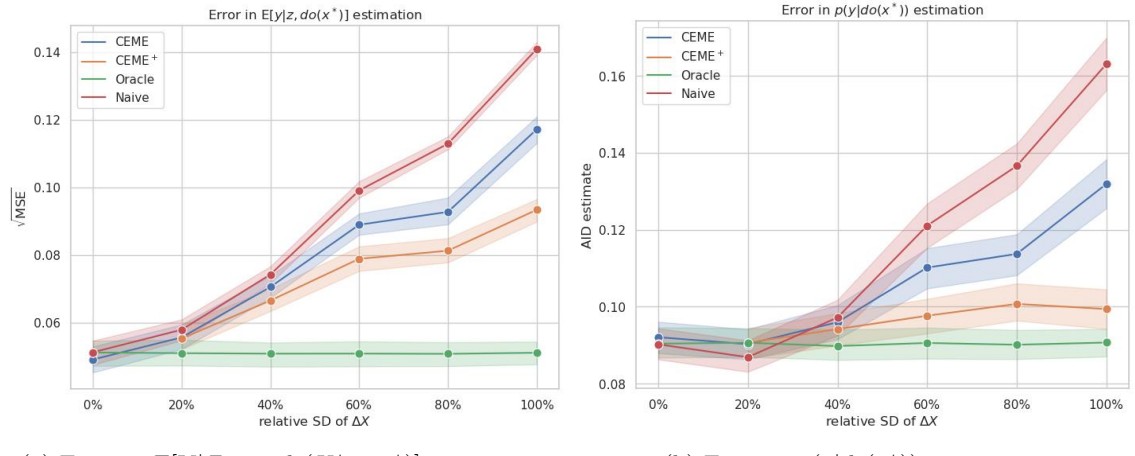

(a) Error in $\mathbb{E}[Y|Z = z, do(X^* = x^*)]$ estimation.

(b) Error in $p(y|do(x^*))$ estimation.

Figure 9: Error in causal effect estimation in the education-wage data experiment. The proposed CEME/CEME$^+$ methods offer improvement compared to the baseline Naive, but not as much as in the synthetic experiment. A likely reason for this is that CEME/CEME$^+$ incorrectly assume a continuous $X^*$, unlike Oracle and Naive. Moreover, the difference between CEME and CEME$^+$ is larger than in the synthetic case.

We run this experiment for the same four algorithms as in the synthetic experiment, defined in Section 2.1, with three hidden layers of width 26 and the ELU activation function. The training procedure and model structures are the same as in Section 3.1 except that now $Z$ is multivariate and hyperparameters are different. Details on the experiment and how the semisynthetic datasets were created are available in Appendix B. The full data of 2990 points is split into 72% of training data, 8% of validation data (used for learning rate annealing and early stopping) and 20% of test data (used for evaluating the models), all amounts rounded to the nearest integer. The code used for data preprocessing and running the experiment is available at `https://github.com/antti-pollanen/ci_noisy _treatment`.

### 3.2.1  Results

The main results of the experiment with education-wage data are presented in Figures 9a, 9b, 10a and 10b. They use the same metrics as the corresponding results for the synthetic data experiment. The points represent median values and the bands represent interquantile ranges. Each data point corresponds to one training run of the model, and the x-axis in each figure indicates which dataset was used (they differ only in the SD of the augmented noise $\Delta X$). There is no entry for CEME$^+$ for the 0% $\Delta X$ SD dataset, because using variational inference for a model with no hidden variables is not useful and is problematic in practice because terms in the ELBO become infinite.

We notice that while CEME and CEME$^+$ still offer a clear improvement over not accounting for measurement at all (Naive), they seem to be further below in performance from Oracle (which acts as a benchmark for optimal, or even beyond-optimal performance for the CEME/CEME$^+$ algorithms). A potential reason for this is that the CEME models, which model the true number of education years as a continuous latent variable, are misspecified unlike Oracle and Naive, which only condition on the number of education years but do not model it (though Naive assumes that there is no measurement error). Moreover, CEME and CEME$^+$ have more parameters than Oracle and Naive; therefore they could be hurt more by the limited dataset size and high-dimensional covariate. The larger difference between CEME and CEME$^+$ than in the synthetic experiment might be because CEME$^+$ avoids some of the detriment from model misspecification by having access to the true standard deviation of $\Delta X$.

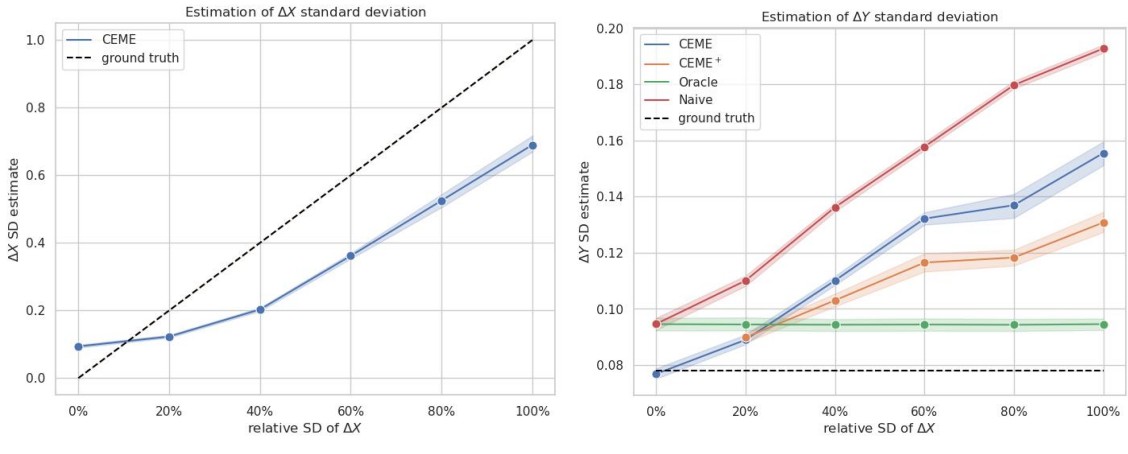

(a) Estimation of $\Delta X$ standard deviation.  (b) Estimation of $\Delta Y$ standard deviation.

Figure 10: Error in the estimation of treatment and outcome noises in the education-wage data experiment. On the left, only CEME is depicted because it is the only method that estimates $\Delta X$ standard deviation. These figures help demonstrate that the proposed CEME/CEME$^+$ offer clear benefit over the baseline Naive even when they are misspecified.

## 4  Conclusion

In this study, we provided a model for causal effect estimation in the presence of treatment noise, with complex nonlinear dependencies, and with no side information or knowledge of the measurement error variance. We confirmed the model's identifiability theoretically and evaluated it experimentally, comparing it to a baseline that does not account for measurement error at all (Naive) and to an upper bound in performance in the form of Oracle. A notable advantage of the model is its flexibility: It offered good performance on a diverse set of synthetic datasets and was useful for correcting for measurement error even on a real-world dataset for which it was clearly misspecified. Our approach is also flexible in a second way: after using amortized variational inference to infer the SCM and the posterior of hidden variables, any interventional or counterfactual distributions may be computed.

A limitation of the CEME/CEME$^+$ methods is their assumption of independent additive noise for both treatment and outcome, which might not always hold Schennach (2016). On the other hand, these assumptions were critical for attaining identifiability without side information; therefore relaxing them would likely mean having to introduce side information to the model Schennach (2020). Another limitation of our study is the lack of comparison with other state-of-the-art methods. However, the authors are not aware of any that exist which estimate the same quantities as our method and are designed for the setting where the treatment is noisy, the regression function does not have any strict parametric form, the model is conditioned on a covariate, and no side information is available. In addition, our study does not consider the robustness of estimation when there is no access to a perfect set of covariates satisfying the backdoor criterion.

Finally, the model is limited by assuming Gaussian distributions in all conditionals. An interesting direction for future research could be to relax this assumption by using flexible distributions, such as normalizing flows. The identifiability result in Schennach & Hu (2013) suggests a model generalized in this way would retain its identifiability. In addition, since it is straightforward to generalize the amortized variational approach for other related latent variable models, an interesting future direction would be to apply the algorithm to models that relax the classical measurement error assumptions Schennach (2016; 2020), for which efficient inference methods are not yet available.

**Broader Impact Statement**

Our method could be used for estimating treatment effects in applications where incorrect inferences might in the worst case lead to severe adverse outcomes, e.g. in healthcare. For this reason it is important to consider the validity of the assumptions of our method in any particular use case as well as to test any implementations carefully before practical deployment.

**Author Contributions**

All authors have accepted responsibility for the entire content of this manuscript and approved to its submission. AP had the main responsibility on all aspects of the work. PM supervised the work and provided the initial research idea. The authors wrote the paper together.

**Acknowledgments**

This work was supported by the Academy of Finland (Flagship programme: Finnish Center for Artificial Intelligence FCAI, and grants 336033, 352986, 358246) and EU (H2020 grant 101016775 and NextGenerationEU).

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

## Appendix

## A    Review of literature on the identifiability of related models

**CEME with categorical variables.** Identifiability is proven by Xia et al. (2020) for a model with the same Bayesian networks as ours (Figure 1) where the treatment $X^*$ and covariate $Z$ are categorical. For this result, misclassification probabilities need to have a certain upper bound to avoid the label switching problem. It is also assumed that the conditional distribution $Y|Z, X^*$ is either a Poisson, normal, gamma, or binomial distribution. This scheme also allows dealing with missing treatment values by defining a missing value as an additional category.

**Measurement error models from a statistical perspective.** There is a wide variety of research on measurement error solely from a statistical (non-causal) perspective, studying assumptions under which measurement error model parameters can be identified, and exploring different practical methods for inferring them. These include the general method of moments (GMM) as well as kernel deconvolution and sieve estimators (Yi et al., 2021; Schennach, 2020). Many of our assumptions are relaxed, such as having Gaussian noise terms, the independence of the measurement error $\Delta X$ or outcome noise $\Delta Y$ of the other variables, and the outcome $Y$ being non-differential, i.e. independent of $\Delta X$. See Schennach (2016; 2020) for a review on this kind of results.

A wide variety of literature also studies measurement error models with additional assumptions compared to ours. These consist mainly of assuming the availability of side information such as repeated measurements or instrumental variables (IVs), or assuming a strict parametric form (e.g. linear or polynomial) for the regression function $\mathbb{E}[Y|Z, X^*]$ (and $\mathbb{E}[X^*|Z]$). See Yi et al. (2021) for a review. The IVs differ from our covariate $Z$ in that they are conditionally independent of $Y$ given $X^*$. On the other hand, Ben-Moshe et al. (2017) consider a covariate $Z$ that directly affects $Y$. However, they assume that the effects of $Z$ and $X^*$ on $Y$ are either decoupled or $\mathbb{E}[Y|Z, X^*]$ has a polynomial parametric form.

**Related latent variable models.** There are also latent variable models studied that resemble the measurement error model. These include the causal effect variational autoencoder (CEVAE), where only a proxy of the confounder is observed (Louizos et al., 2017; Rissanen & Marttinen, 2021) and nonlinear independent component analysis (Khemakhem et al., 2020). Also, Schennach (2016; 2020) consider the identification of latent variable models more generally.

## B    Experiment details

Amortized variational inference is prone to getting stuck at local optima, for example, in the case of the so-called posterior collapse (Kingma & Welling, 2019). To counter this, we start training in both experiments with an increased but gradually annealed weight for the ELBO terms $\log q_\phi(x^*_{i,j}|s_i)$ and $\log p_\theta(x_i|x^*_{i,j})$, which causes the model to initially predict posterior $x^*$ close to $x$, to ensure that the posterior does not get stuck at the prior. A related approach was taken by Hu et al. (2022), where the model was pre-trained assuming no measurement error.

### B.1    Synthetic experiment

In the synthetic experiment data generation, the functions $\mu_{X^*}$, $\sigma_{X^*}$ are only approximately sampled from a GP to save in computation cost, as with exact values the size of the GP kernel scales proportionally to the square of the number of data points, and the matrix operations performed with it scale even worse. Thus, we sample only $K = 1000$ points from the actual Gaussian processes corresponding to each of the functions $\mu_{X^*}$, $\sigma_{X^*}$ and $\mu_Y$. These functions are then defined as the posterior mean of the corresponding GP, given that the $K$ points have been observed. The points are evenly spaced such that the minimum is the smallest value in the actual generated data minus one quarter of the distance between the smallest and largest values in the actual data. Similarly, the maximum is the largest value in the actual data plus the distance between the smallest and largest values in the actual data. This is achieved by first generating the actual values $z_{i_{i=1..N}}$, then sampling the $K$ points for $\mu_{X^*}$, and $\sigma_{X^*}$, then generating $x^*_{i=1..N}$ and then finally sampling

the $K$ points for $\mu_Y$.) For $\mu_Y$, as it has two arguments, we used $K = 31^2 = 961$ points arranged in a two-dimensional grid.

When training the model, learning rate annealing is used, which means that if there have been a set number of epochs without improvement in the validation score, the learning rate will be multiplied by a set learning rate reduction factor. The training is stopped when a set number of epochs has passed without the validation score improving.

The hyperparameter values used are listed in Table 1. They were optimized using a random parameter search. We use 8000 data points as the validation set used for annealing learning rate and for early stopping. Unless otherwise stated, the same hyperparameter value was used for all algorithms and datasets. From all the 43200 runs in the synthetic experiment, only 33 runs crashed, yielding no result.

Table 1: Hyperparameter values used in the synthetic experiment.

| Hyperparameter | Value |
|---|---|
| Number of hidden layers in each network (fully connected) | 3 |
| Width of each hidden layer | 20 |
| Activation function | ELU |
| Weight decay | 0 |
| Number of importance samples | 32 |
| Initial weight of $\log q_\phi(x^*_{i,j}|s_i)$ and $\log p_\theta(x_i|x^*_{i,j})$ terms (see Section 2.2.1) | 4 |
| Number of epochs to anneal above weight to 1 | 10 |
| Learning rate reducer patience | 30 |
| Learning rate reduction factor | 0.1 |
| Early stopping patience in epochs | 40 |
| Adam $\beta_1$ | 0.9 |
| Adam $\beta_2$ | 0.97 |
| Batch size used for CEME/CEME$^+$ when training dataset size is 16000 | 256 |
| Batch size otherwise | 64 |
| Learning rate for CEME/CEME$^+$ with training dataset sizes 1000 and 4000 | 0.003 |
| Learning rate for CEME/CEME$^+$ with training dataset size 16000 | 0.01 |
| Learning rate for Oracle and Naive | 0.001 |

## B.2 Experiment with education-wage data

For the education-wage dataset by Card (1995), the covariates used were personal identifier, whether the person lived near a 2 year college in 1966, whether the person lived near a 4 year college in 1966, age, whether the person lived with both parents or only mother or with step parents at the age of 14, several variables on which region of USA the person lived in, whether the person lived in a metropolitan area (SMSA) in 1966 and/or 1967, whether the person was enrolled in a school in 1976, whether the person was married in 1976, and whether the person had access to a library card at the age of 14.

The semi-synthetic dataset used in our experiments is obtained from the original dataset by Card (1995) as follows: We exclude multiple covariates present in the original dataset that correlate heavily with the number of education years and would thus make measurement error correction much less useful. These include the number of schooling years of the mother and father, their intelligence quotients, Knowledge of the World of Work (KWW) score, work experience in years, its square, work experience years divided into bins, and wage. We also drop NLS sampling weight. Missing values are handled by dropping all data items that contain them. This reduces the size of the dataset from 3010 to 2990.

Then, both $X^*$ (number of education years) and all covariates in $Z$ are scaled to have a zero mean and unit variance. The noisy treatment $X$ is obtained by adding a normally distributed additive noise $\Delta X$ to $X^*$. Six separate datasets are created, each corresponding to a different level of SD of $\Delta X$. The levels are proportional to the SD of $X^*$, and are 0%, 20%, 40%, 60%, 80% and 100%.

A synthetic outcome $Y$ is generated as follows based on the true value of the logarithm of wage: First, we train a neural network (five hidden layers of size 30, weight decay 0.01 and ELU activation function) to predict the logarithm of wage (scaled to have zero mean and unit variance). We then use the trained neural network as the function $\mu_Y[Z, X^*] = \mathbb{E}[Y|Z, X^*]$. To obtain $Y$, we add to this expectation a Gaussian noise $\Delta Y$ whose SD is set to 10% of the true SD estimated with training data for the $\mu_Y[Z, X^*]$ neural network. This neural network was trained on all the data, but was regularized so as not to overfit to a meaningful extent. The SD is only 10% of the estimated true SD, because otherwise the predictions are so inaccurate to begin with that better handling of measurement error has little potential to improve the results. The synthetic $Y$ is also used as the ground truth for $\mathbb{E}[Y|Z, X^*]$.

The hyperparameter values used are listed in Table 2. The hyperparameters are shared by all algorithms and were optimized using a random search. A separate search was conducted for each type of algorithm, but the optima were close enough that for simplicity, the same values could be chosen for each algorithm.

Table 2: Hyperparameter values used in the experiment with education-wage data.

| Hyperparameter | Value |
|---|---:|
| Number of hidden layers in each network (fully connected) | 3 |
| Width of each hidden layer | 26 |
| Activation function | ELU |
| Batch size | 32 |
| Learning rate | 0.001 |
| Weight decay | 0.001 |
| Number of importance samples | 32 |
| Initial weight of $\log q_\phi(x^*_{i,j}|s_i)$ and $\log p_\theta(x_i|x^*_{i,j})$ terms (see Section 2.2.1) | 8 |
| Number of epochs to anneal above weight to 1 | 5 |
| Learning rate reducer patience | 25 |
| Learning rate reduction factor | 0.1 |
| Early stopping patience in epochs | 45 |
| Adam $\beta_1$ | 0.9 |
| Adam $\beta_2$ | 0.97 |

## C  Identification theorem used to prove Proposition 1

For completeness, we include below Theorem 1 by Schennach & Hu (2013) (repeated mostly word-for-word):

**Definition 2.** We say that a random variable $r$ has an *F factor* if $r$ can be written as the sum of two independent random variables (which may be degenerated), one of which has the distribution $F$.

*Model 1.* Let $y$, $x$, $x^*$, $\Delta x$, $\Delta y$ be scalar real-valued random variables related through

$$y = g(x^*) + \Delta y \tag{13}$$
$$x = x^* + \Delta x, \tag{14}$$

and $y$ are observed while all remaining variables are not and satisfy the following assumptions:

*Assumption 1.* The variables $x^*$, $\Delta x$, $\Delta y$, are mutually independent, $\mathbb{E}[\Delta x] = 0$, and $E[\Delta y] = 0$ (with $\mathbb{E}[|\Delta x|] < \infty$ and $\mathbb{E}[|\Delta y|] < \infty$).

*Assumption 2.* $E[e^{i\xi\Delta x}]$ and $E[e^{i\gamma\Delta y}]$ do not vanish for any $\xi, \gamma \in \mathbb{R}$, where $i = \sqrt{-1}$.

*Assumption 3.* (i) $E[e^{i\xi x^*}] \neq 0$ for all $\xi$ in a dense subset of $\mathbb{R}$ and (ii) $E[e^{i\gamma g(x^*)}] \neq 0$ for all $\gamma$ in a dense subset of $\mathbb{R}$ (which may be different than in (i)).

*Assumption 4.* The distribution of $x^*$ admits a uniformly bounded density $f_{x^*}(x^*)$ with respect to the Lebesgue measure that is supported on an interval (which may be infinite).

*Assumption 5.* The regression function $g(x^*)$ is continuously differentiable over the interior of the support of $x^*$.

*Assumption 6.* The set $\chi = \{x^* : g'(x^*) = 0\}$ has at most a finite number of elements $x_1^*, ..., x_m^*$. If $\chi$ is nonempty, $f_{x^*}(x^*)$ is continuous and nonvanishing in a neighborhood of each $x_k^*$, $k = 1, ..., m$.

**Theorem 1.** *Let Assumptions 1-6 hold. Then there are three mutually exclusive cases:*

1. *$g(x^*)$ is not of the form*

$$g(x^*) = a + b \ln(e^{cx^*} + d) \tag{15}$$

   *for some constants $a, b, c, d \in \mathbb{R}$. Then, $f_{x^*}(x^*)$ and $g(x^*)$ (over the support of $f_{x^*}(x^*)$) and the distributions of $\Delta x$ and $\Delta y$ in Model 1 are identified.*

2. *$g(x^*)$ is of the form (15) with $d > 0$ (A case where $d < 0$ can be converted into a case with $d > 0$ by permuting the roles of $x$ and $y$). Then, neither $f_{x^*}(x^*)$ nor $g(x^*)$ in Model 1 are identified iff $x^*$ has a density of the form*

$$f_{x^*}(x^*) = A \exp(-Be^{Cx^*} + CDx^*)(e^{Cx^*} + E)^{-F}, \tag{16}$$

   *with $c \in \mathbb{R}$, $A, B, D, E, F \in [0, \infty]$ and $\Delta y$ has a Type I extreme value factor (whose density has the form $f_u(u) = K_1, \exp(K_2 \exp(K_3 u) + K_4 u)$ for some $K_1, K_2, K_3, K_4 \in \mathbb{R}$).*

3. *$g(x^*)$ is linear (i.e., of the form (15) with $d = 0$). Then, neither $f_{x^*}(x^*)$ nor $g(x^*)$ in Model 1 are identified iff $x^*$ is normally distributed and either $\Delta x$ or $\Delta y$ has a normal factor.*

## D  Proof of Proposition 1

*Proof.* First, consider the measurement error model defined in Equations (1)–(5) conditioned on a specific value $Z = z$, effectively removing $Z$ as a variable. We show that this model, called the *restricted model*, as opposed to the original *full model*, satisfies the assumptions in Theorem 1, which thus determines when the restricted model is identifiable. First, the restricted model satisfies Equations (13) and (14). Assumption 1 follows directly from the definition of the model, as $|\Delta X|$ and $|\Delta Y|$ follow the half-normal distribution, which has the known finite expectation $\sigma \sqrt{2}/\sqrt{\pi}$. Assumption 2 is satisfied because $\Delta X$ and $\Delta Y$ are Gaussian so their characteristic functions have the known form $\exp(i\mu t - \sigma^2 t^2/2)$ and thus do not vanish for any $t \in \mathbb{R}$ ($t$ is denoted by $\xi$ or $\gamma$ in Assumption 2).

Assumption 3 of Theorem 1 is not needed because it is in its proof by Schennach & Hu (2013) only used to find the distributions of the errors $\Delta X$ and $\Delta Y$ given that the density $f_{x^*}(x^*)$ and regression function $g(x^*)$ are known. However, in our case we already know by assumption that the error distributions are Gaussian with a zero mean, and moreover, we can find their standard deviation from

$$\text{Var}[Y] = \text{Var}[\mu_Y(z, X^*)] + \text{Var}[\Delta Y]$$

and

$$\text{Var}[X] = \text{Var}[X^*] + \text{Var}[\Delta X],$$

which hold because of the independence of $\mu_Y(z, X^*)$ and $\Delta Y$ as well as $X^*$ and $\Delta X$, respectively. Assumption 4 is satisfied because $X^*|Z = z$ is normally distributed and thus admits a uniformly bounded density w.r.t. the Lebesgue measure that is supported everywhere. Assumption 5 is the same as assumption 1 of our Proposition 1. Assumption 6 follows from assumption 2 of Proposition 1 since the density of $X^*|Z = z$ is continuous and nonvanishing everywhere.

With the assumptions of Theorem 1 satisfied, we check what the cases 1-3 therein imply for our restricted model. From case 1 we obtain that it is identified except when $\mu_Y(z, x^*)$ as a function of $x^*$ is of the form $a + b \ln(e^{cx^*} + d)$.

Case 2 considers the case when $\mu_Y(z, x^*)$ as a function of $x^*$ is of the form $a + b\ln(e^{cx^*} + d)$ with $d > 0$. (Having $d < 0$ is impossible as we assume the function $g$ is defined on $\mathbb{R}$). In this case, our model is identifiable, as $X^*$ is Gaussian and thus its density is not of the form

$$f_{x^*}(x^*) = A\exp(-Be^{Cx^*} + CDx^*)(e^{Cx^*} + E)^{-F} \tag{17}$$

with $C \in \mathbb{R}$ and $A, B, D, E, F \in [0, \infty)$. We see this by taking a logarithm of both the Gaussian density and the density in Equation (17) and noting that the first is a second-degree polynomial, but the latter is a sum without a second-degree term, so they are not equal. Thus, our model is identifiable in this case.

The remaining case is for when $\mu_Y(z, x^*)$ is linear in $x^*$. In this case, the restricted model is not identified as $x^*$ is normally distributed and both $\Delta x$ and $\Delta y$ have normal factors as they are normal themselves.

Next, we prove the identifiability of the full model, i.e. where $z$ may take any value in its range. We start by assuming in accordance with Definition 1 that two conditional observed distributions from the model match for every $z$:

$$\forall z, x, y: \quad p_\theta(x, y|z) = p_{\theta'}(x, y|z). \tag{18}$$

Now based on assumption 3 of Proposition 1, there exists $\bar{z}$ for which $\mu_Y(\bar{z}, x^*)$ is not linear in $x^*$. From (18) we obtain that

$$\forall x, y: \quad p_\theta(x, y|\bar{z}) = p_{\theta'}(x, y|\bar{z}),$$

which in turn implies

$$\tau = \tau'$$
$$\sigma = \sigma'$$
$$\mu_{X^*}(\bar{z}) = \mu'_{X^*}(\bar{z}) \tag{19}$$
$$\sigma_{X^*}(\bar{z}) = \sigma'_{X^*}(\bar{z})$$
$$\forall x^* \in \mathbb{R}: \quad \mu_Y(\bar{z}, x^*) = \mu'_Y(\bar{z}, x^*)$$

by using Definition 1 on the restricted model, which is identified according to the first part of the proof.

Similarly, we obtain the equalities in (19) for every $\bar{z}$ for which $\mu_Y(\bar{z}, x^*)$ is not linear in $x^*$. Thus, for the full equality of the models (i.e. for $\theta = \theta'$), it remains to be shown that we obtain $\mu_{X^*}(z) = \mu'_{X^*}(z)$, $\sigma'_{X^*}(z) = \sigma_{X^*}(z)$ and $\mu_Y(z, x^*) = \mu'_Y(z, x^*)$ also for those $z$ for which $\mu_Y(z, x^*)$ is linear in $x^*$. Noting that we already know that $\tau = \tau'$ and $\sigma = \sigma'$, we obtain for such $z$ a linear-Gaussian measurement error model with known measurement error, which is known to be identifiable (see e.g. Gustafson (2003)). Thus $\theta = \theta'$ and the proof is complete. $\qquad\square$

