# OpenReview forum: "Identifiable Causal Inference with Noisy Treatment and No Side Information"
_TMLR — Accepted by TMLR_

### Review · Reviewer_y5Kg · 2024-05-29

**Summary Of Contributions:**

This paper aims to carry out causal inference in settings where the observed variables are subject to noise. They aim to account for this noise _without_ using extra _side_ information. The work solves for a causal graph using amortised variational inference, and test their method on synthetic data, as well as on education wage data.

**Audience:**

Yes

**Broader Impact Concerns:**

Since the applications of this paper are medical and economical, I wonder if a brief Broader Impact statement would be good.

**Claims And Evidence:**

Yes

**Requested Changes:**

- Could an example be given in the introduction for where the scenario in which the work in the paper is relevant? The paper mentions _complicated by variables being subject to noise, e.g., inaccurate measurement, clerical error, or self-reporting (often called misclassification for categorical variables)._, etc, but I think a grounded example could help. I think this would go well when describing Figure 2, where you can given an example of a dataset / scenario where the causal graph of Figure 2 applies.

- Page 1: _'...how a **cause** variable X, called treatment...'_: is this a typo? Should it be **causal** variable?

- Page 1 and 2: '_Figure 1 presents an example of this skewing effect_':  Could you explain more about how Figure 1 shows a skewing effect in the text where you say that Figure 1 presents a skewing effect?

- Figure 1:  I am confused about how Figure 1 is generated. Could this be explained in more detail? What do you mean by a Ground Truth _regression_ function? Is this the function from which the data is generated? If so, what is the difference between the accurate data and the noisy data?

- Page 3: What is the difference between $\mu_y(Z, X^*)$ and $\mu_y(z, x^*)$? If there is a difference, could this be explained?

- Page 3: I am confused by the notation of $\Delta X$, etc. Why not just use $U_X$, etc?

- Algorithm 1: In step 2, what does it mean to sample approximately from a GP(0, K)?

- Page 10: _To evaluate our methods, we need to know the ground truth causal effect, _, could you explain this further?

- Figures: I think that the figure captions should have sentences about what the key takeaway of each figure is.

- Appendixes: I think it would be clearer if the neural network architecture details are added to a table, or are a part of Tables 1 and 2, or are highlighted in some, so that they are easier to parse quickly.

**Strengths And Weaknesses:**

### Strengths
I think that the paper is well written with generally clear exposition. However, there are some notation in equations that could be explained better.

- It is good that the key hyper parameters of the training procedures is mentioned.

### Weaknesses
See Requested Changes.

---

> ### Author Response · Authors · 2024-07-02
> **Response to reviewer feedback**
>
> Reply to the first requested change "Could an example be given in the introduction for where ...":  We agree this would be a good addition. A good example in the context of healthcare would be the effect of blood pressure on some continuous health outcome, e.g. arterial stiffness as measured by pulse wave velocity (the speed of arterial pressure waves traveling along the aorta and large arteries), kidney function as measured by glomerular filtration rate (GFR), or the concentration of albumin in urine. The accurate treatment X* is a sliding window average of the instantaneous blood pressure X, which varies highly from moment to moment, and is assumed to have been measured only once per subject.
>
> We have added this example to the introduction.
>
> > Page 1: '...how a cause variable X, called treatment...': is this a typo? Should it be causal variable?
>
> Cause variable here refers to cause as in cause and effect. However, we agree that this sentence is clumsy and changed it from “Causal inference deals with how a cause variable $X$, called treatment, causally affects an outcome $Y$” to “Causal inference deals with how a treatment variable $X$ causally affects an outcome $Y$”.
>
>
> > Page 1 and 2: 'Figure 1 presents an example of this skewing effect': Could you explain more about how Figure 1 shows a skewing effect in the text where you say that Figure 1 presents a skewing effect?
>
> We have clarified this in the revision as you suggest.
>
> > Figure 1: I am confused about how Figure 1 is generated. Could this be explained in more detail? What do you mean by a Ground Truth regression function? Is this the function from which the data is generated? If so, what is the difference between the accurate data and the noisy data?
>
> We agree this should be made more clear and have done so in the revision by clarifying the caption and the legend of the figure. Yes, the ground truth is the mean function of the data generating process. Accurate and noisy data are the same data points but such that there is measurement error in the treatment variable X* in the noisy data (the y-values are the same, and noisy, in both datasets). It can be seen that our method “CEME” fits the noise-free data (even if that is not seen by any method) whereas the “naive” method fits the noisy data and can not estimate the true regression function accurately.
>
> Also, the usage of the term noisy data was ambiguous, since the true values of the variables $X^*$ and $Y$ contain inherent noise in both cases. Rather, the difference between the two data is whether measurement error for $X^*$ is included. We have changed the terminology accordingly in the revision.
>
> > Page 3: What is the difference between $\mu_y(Z, X^*)$ and $\mu_y(z, x^*)$? If there is a difference, could this be explained?
>
> We agree this could be clarified, and we added the following to the revision: “Note that by uppercase letters we denote random variables and by lowercase letters their specific realized values. Thus $\mu_Y(z,x^*)$ refers to the value of $\mu_Y$ with argument values $z$ and $x^*$, while $\mu_Y(Z,X^*)$ denotes the random variable that is obtained by applying $\mu_Y$ to the random variables $Z$ and $X^*$.”
>
>
>
> > Page 3: I am confused by the notation of $\Delta X$, etc. Why not just use $U_X$, etc?
>
> The advantage of using $\Delta X$ is that it is used also by Schennach in her earlier works on which our work is based. In this way the notation in our paper and proofs is consistent with the earlier works, which we believe helps to avoid confusion caused by different notations.
>
>
> > Algorithm 1: In step 2, what does it mean to sample approximately from a GP(0, K)?
>
> To simplify the algorithm description, we remove the word “approximately”.
>
> We edited the text to the following: “To avoid excessive computational cost, the functions $\mu_{X^*}(z)$, $\sigma_{X^*}(z)$ and $\mu_Y(z, x^*)$ are only approximations of true samples from the GPs (steps 2 and 4 in Algorithm 1), as described in Appendix B along with other experiment details.”
>
>
> >  Page 10: _To evaluate our methods, we need to know the ground truth causal effect, _, could you explain this further?
>
> We agree this could be clarified, and added an explanation to the revision:
>
>
> > Figures: I think that the figure captions should have sentences about what the key takeaway of each figure is.
>
> We implemented this in the revision.
>
>
> > Appendixes: I think it would be clearer if the neural network architecture details are added to a table, or are a part of Tables 1 and 2, or are highlighted in some, so that they are easier to parse quickly.
>
> Good point! In the revision we added the neural network architecture details (number of hidden layers, hidden layer width and activation function) to Tables 1 and 2 in the appendix, as suggested.
>
> > Since the applications of this paper are medical and economical, I wonder if a brief Broader Impact statement would be good.
>
> We decided to add a Broader Impact statement in the revision.

---

### Review · Reviewer_qb2H · 2024-06-11

**Summary Of Contributions:**

This paper focus on the scenario with noisy treatment, nonlinear dependencies, and no side information. It not only provides identifiability analysis but also develops a method for inferring causal effects. Extensive experimental results demonstrate the effectiveness of their proposed method.

**Audience:**

Yes

**Claims And Evidence:**

Yes

**Requested Changes:**

1. In line 2 of the second paragraph of Section 1, "Unaccounted" should be replaced with an adverb.

2. In the fourth paragraph of Section 1, they authors claim that "The assumption of independent exogenous variables implies that there is no unobserved confounding", which is incorrect. The author should implicitly claim that they employ the causal sufficiency assumption, violation of this assumption is not incompatible with independent exogenous variables.

3. In page 3, the authors claim that "... using so called side-information, such as a known error mechanism, ...". However, they assume independent and zero-mean additive errors, (i.e. known error mechanism) and claim that they need no side information. This is quite confusing.

4. In Section 2.2.1, "The standard ELBO corresponds to $k=1$" should be replaced with "The standard ELBO corresponds to $K=1$".

**Strengths And Weaknesses:**

Strengths:

1. This work investigate a challenging setting with measurement treatment, complex nonlinear dependencies, and no side information.

2. This work provides extensive empirical results to demonstrate the effectiveness of their proposed method.

Weaknesses:

1. The contribution of this paper is marginal. The setting investigated in this paper is similar to Gao et al. (2024), both its inference method and identifiability analysis builds closely upon previous works (Zhang et al., 2019) and (Schennach & Hu (2013)). Besides, all experiments employ Guassian noises, I'm not sure whether the proposed method can handle non-Gaussian cases properly.

2. The organization of this paper is confusing. The authors present too many details in the introduction, which should be deferred to "Related Work" section or "Preliminaries" section. In the introduction, they only need to formulate the setting they investigated clearly and emphasize their main contributions.

3. The writing of this paper should be improved, please see more details in "requested changes".

---

> ### Author Response · Authors · 2024-07-02
> **Response to reviewer feedback**
>
> > The contribution of this paper is marginal. The setting investigated in this paper is similar to Gao et al. (2024), both its inference method and identifiability analysis builds closely upon previous works (Zhang et al., 2019) and (Schennach & Hu (2013)). Besides, all experiments employ Guassian noises, I'm not sure whether the proposed method can handle non-Gaussian cases properly.
>
> Thank you for this comment. Below we try to clarify the relation of the mentioned works to our paper. We will include similar clarifications to the article.
>
> Gao et al. (2024) is independent of and parallel with our work, and actually appeared online after our paper. A strength of our work is an explicit proof of the identifiability.
>
> We openly acknowledge that the proof here is an extension of the result by Schennach et al. (2013), which we have extended by including covariates and adjusted to the specific assumptions of our model.
>
> Our VAE model assumes Gaussian noise. A VAE with some other noise distribution (e.g. a normalizing flow) should still be identifiable based on the work by Schennah & Hu, but a more detailed  investigation of that is outside the scope of our paper.
>
> Zhang et al. 2019 considers VI and VAE models in general and does not focus on estimating measurement error models or causal inference.
>
>
> > The organization of this paper is confusing. The authors present too many details in the introduction, which should be deferred to "Related Work" section or "Preliminaries" section. In the introduction, they only need to formulate the setting they investigated clearly and emphasize their main contributions.
>
> We agree that the introduction section is now too heavy, and moved the discussion about the SCMs and identifiability as well as the model definition to Section 2.1 “Models for causal estimation”. We decided for now against a preliminaries section since the preliminary material on SCMs and identifiability that would belong there is just two paragraphs. Also, we decided for now against a related work section, since the references help to clarify what our contribution is relative to the existing literature.
>
> If you would still like to see a “Related work” or “preliminaries” section, or any other changes, we are happy to respond to additional feedback and make the necessary changes.
>
>
>
> > In line 2 of the second paragraph of Section 1, "Unaccounted" should be replaced with an adverb.
>
> We replaced “Unaccounted, …” with “If not accounted for, …”.
>
> > In the fourth paragraph of Section 1, they authors claim that "The assumption of independent exogenous variables implies that there is no unobserved confounding", which is incorrect. The author should implicitly claim that they employ the causal sufficiency assumption, violation of this assumption is not incompatible with independent exogenous variables.
>
> We indeed agree that our current formulation is inaccurate. What we meant was that we assumed a certain SCM, which did not include unobserved confounding, which implies causal sufficiency. We have modified the sentence to: “The SCM with independent noise terms and no hidden confounders implies the common assumption of causal sufficiency.”
>
>
> > In page 3, the authors claim that "... using so called side-information, such as a known error mechanism, ...". However, they assume independent and zero-mean additive errors, (i.e. known error mechanism) and claim that they need no side information. This is quite confusing.
>
> We agree this requires clarification. Indeed, it seems that in its common usage “side-information” refers to additional per-datapoint information, which includes repeated measurements or instruments. Additionally, it seems reasonable to include a gold-standard sample of accurate measurements under “side information”, since in this case the accurate values are the additional per-datapoint information.
>
> Independent and zero-mean additive errors could be seen as a lesser requirement since it could reasonably in some cases be assumed a-priori without additional measurements. These are also very common assumptions in the literature, together called “strongly classical measurement error”.
>
> Here with “known error mechanism” we meant that the variance of the noise would be known, which is another common assumption to constrain the model and make it identifiable, and our method does not require this information either. We will replace “known error mechanism” with “known error variance” in the article.
>
>
> > In Section 2.2.1, "The standard ELBO corresponds to $k=1$" should be replaced with "The standard ELBO corresponds to $K=1$".
>
> Thank you for pointing this out, we made this change in the revision.

---

### Review · Reviewer_La5Z · 2024-06-24

**Summary Of Contributions:**

This paper studies causal inference under inaccurate observation variables and without side information. To measure the error, a new causal model is proposed. The authors propose a method to train the model and analyze its identifiability. Experiments on synthetic and real-world datasets are conducted to verify the method.

**Audience:**

Yes

**Broader Impact Concerns:**

This work does not present direct ethical concerns or adverse broader societal impacts.

**Claims And Evidence:**

Yes

**Requested Changes:**

1. In subfigure (a), Figure 10, the experiment results for other methods are missing.
2. It is better to denote which is the proposed method (CEME and CEME+), which is the baseline (CE-simple), and which is the goal (CE-oracle).

**Strengths And Weaknesses:**

Strengths:

1. The setting in this paper, causal inference under the inaccurate observation variables and without side information, is interesting.
2. An identifiability analysis is provided. Sufficient experiments are conducted to verify the methods.

Weaknesses:

1. The organization and writing of this paper is poor.
2. In Figure 1, is only the noisy data available for the model? The accurate data also contains noise.
3. In equations 5-6, why do the authors assume that the mean of the Gaussian distribution is zero, not a learnable variable?

---

> ### Author Response · Authors · 2024-07-02
> **Response to reviewer feedback**
>
> > The organization and writing of this paper is poor.
>
> The other reviewer (qb2H) also recommended improving the organization by making the introduction more focused by moving some material from the Introduction to Related Work and/or Preliminaries. In the revision we have moved material from the introduction to Section 2.1 (Models for causal estimation.). In detail, we also put there the material on SCMs and identifiability that could be seen as preliminaries, since it consists only of two paragraphs. We decided for now against a related work section, since the references help to clarify what our contribution is relative to the existing literature.
>
> If you have any additional suggestions we are happy to take those into account in any additional revision.
>
> > In Figure 1, is only the noisy data available for the model? The accurate data also contains noise.
>
> We replaced “accurate data” with “accurate X*” and “noisy data” with “error in X*” in the legend, and in general reworked the explanation of the example. Confusion was perhaps caused by us not distinguishing between noise and measurement error: even the true values of the variables Y and X* contain inherent noise, but only for X* is additional measurement error explicitly modeled.
>
> > In equations 5-6, why do the authors assume that the mean of the Gaussian distribution is zero, not a learnable variable?
>
> Making some assumption on the location of the error distributions $\Delta X$ and $\Delta Y$ (Equations 5-6) is critical for identifiability.  We have clarified these points in the paper in Section 2.1.
>
> > In subfigure (a), Figure 10, the experiment results for other methods are missing.
>
> The other methods are not included because they do not include estimation of the standard deviation of $\Delta X$. In the revision we have clarified this in the figure caption.
>
>
> > It is better to denote which is the proposed method (CEME and CEME+), which is the baseline (CE-simple), and which is the goal (CE-oracle)
>
> We have renamed “CE-oracle” to “Oracle” and “CE-simple” to “Naive”, to emphasize their role as reference points for our proposed variants CEME and CEME+. In addition, we have clarified this in the end of Section 2.1 where the different models are introduced.

---

### Decision · Action_Editor_sXFK · 2024-08-15

**Recommendation:** Accept as is

**Comment:**

This paper contributes to the field of causal inference. It specifically targets complex scenarios with noisy treatments, nonlinear dependencies, and absence of side information.

During the review process, concerns were initially raised regarding the organization and clarity of the manuscript. Following the authors' revisions and responses, these concerns have been adequately addressed, and the claims are now well-supported. Reviewers have appreciated these revisions, showing a consensus towards acceptance of the paper. The remaining point of discussion pertains to the technical contributions, which is not a major concern according to TMLR's evaluation criteria. Given the above points, and aligning with the reviewers' final leanings, I recommend acceptance of this paper.

**Audience:**

The paper's focus on causal inference for noisy treatments and nonlinear dependencies addresses significant challenges in real-world scenarios. This topic is likely to attract the interest of researchers in causal analysis and potentially those in broader areas who deal with similar complexities in data.

**Claims And Evidence:**

The evidence aligns well with TMLR's standards. All reviewers acknowledge that the major claims are well supported by the evidence presented in the manuscript. The authors have effectively demonstrated the proposed method's capabilities through rigorous identifiability analyses and extensive experimental results.